# Characterization of Probiotic Properties and Whole-Genome Analysis of *Lactobacillus johnsonii* N5 and N7 Isolated from Swine

**DOI:** 10.3390/microorganisms12040672

**Published:** 2024-03-28

**Authors:** Kun Wang, Yu Wang, Lifang Gu, Jinyan Yu, Qianwen Liu, Ruiqi Zhang, Guixin Liang, Huan Chen, Fang Gu, Haoyu Liu, Xin’an Jiao, Yunzeng Zhang

**Affiliations:** 1Jiangsu Co-Innovation Center for Prevention and Control of Important Animal Infectious Diseases and Zoonoses, Yangzhou University, Yangzhou 225009, China; 13165438089@163.com (K.W.); wy8042wy@163.com (Y.W.); gulifang1997@163.com (L.G.); eve_421@163.com (J.Y.); liuqianwen1177@163.com (Q.L.); rickyuuz@163.com (R.Z.); yigemorning@163.com (G.L.); ch1468599954@163.com (H.C.); 2Jiangsu Key Laboratory of Zoonosis, Yangzhou University, Yangzhou 225009, China; 3Joint International Research Laboratory of Agriculture and Agri-product Safety of the Ministry of Education, Yangzhou University, Yangzhou 225009, China; fanggu1011@163.com (F.G.); haoyu.liu@yzu.edu.cn (H.L.); 4Key Laboratory of Prevention and Control of Biological Hazard Factors (Animal Origin) for Agrifood Safety and Quality, Ministry of Agriculture of China, Yangzhou University, Yangzhou 225009, China; 5College of Animal Science and Technology, Yangzhou University, Yangzhou 225009, China

**Keywords:** *Lactobacillus johnsonii*, probiotic properties, comparative genome analysis, complete genome sequence

## Abstract

In our previous microbiome profiling analysis, *Lactobacillus* (*L.*) *johnsonii* was suggested to contribute to resistance against chronic heat stress-induced diarrhea in weaned piglets. Forty-nine *L. johnsonii* strains were isolated from these heat stress-resistant piglets, and their probiotic properties were assessed. Strains N5 and N7 exhibited a high survival rate in acidic and bile environments, along with an antagonistic effect against *Salmonella*. To identify genes potentially involved in these observed probiotic properties, the complete genome sequences of N5 and N7 were determined using a combination of Illumina and nanopore sequencing. The genomes of strains N5 and N7 were found to be highly conserved, with two N5-specific and four N7-specific genes identified. Multiple genes involved in gastrointestinal environment adaptation and probiotic properties, including acidic and bile stress tolerance, anti-inflammation, CAZymes, and utilization and biosynthesis of carbohydrate compounds, were identified in both genomes. Comparative genome analysis of the two genomes and 17 available complete *L. johnsonii* genomes revealed 101 genes specifically harbored by strains N5 and N7, several of which were implicated in potential probiotic properties. Overall, this study provides novel insights into the genetic basis of niche adaptation and probiotic properties, as well as the genome diversity of *L. johnsonii.*

## 1. Introduction

Lactic acid bacteria (LAB) are generally defined as a group of Gram-positive, non-spore-forming bacteria that are capable of producing lactic acid and exhibit negative catalase activity [1]. Numerous members of the LAB exhibit probiotic characteristics, with a notable emphasis on those affiliated with the genus *Lactobacillus* (taxonomy lineage *Firmicutes*, *Bacilli*, *Lactobacillales*, *Lactobacillaceae*) constituting a core group in this context. Probiotic members of the *Lactobacillus* genus can colonize the intestine and maintain the balance of intestinal microbiota, which includes inhibiting the growth of harmful bacteria in the intestine through various mechanisms such as producing organic acids, bacteriocin, and other antibacterial substances [2]. Furthermore, they exhibit a close association with the cells located in the lower regions of the intestinal mucosa and then benefit the host by participating in immune cell regulation and activation [3]. 

Among the species affiliated with the genus *Lactobacillus*, *L. johnsonii* is mainly distributed in the gastrointestinal tract of both humans and non-human animals [4]. *L. johnsonii* has been isolated from the intestines of piglets [5], dogs [6], chickens [7], calves [8], mice [9], bees [10], and humans [11]. It usually exhibits resistance against the corrosive effects of gastric acid and bile compounds, enabling it to traverse the gastrointestinal tract and adhere effectively [12]. *L. johnsonii* is regarded as a safe probiotic microorganism, holding a Generally Recognized as Safe (GRAS) status, and is included in the Qualified Presumption of Safety (QPS) list [13]. Many members of *L. johnsonii* exhibit several probiotic characteristics, including the maintenance of metabolic stability in the gastrointestinal tract, inhibition of in vivo growth and proliferation of pathogenic bacteria, and enhancement of animal growth performance. Given these probiotic characteristics, several *L. johnsonii* probiotic strains have been extensively utilized as probiotics in feed additives and probiotic preparations [14,15]. 

With the recent advancements in high-throughput sequencing technologies, notably long-read sequencing technologies, it has become feasible to acquire complete genome sequences of bacterial strains [16]. This capability enhances the precise and confident identification of genes associated with probiotic properties and other traits through genome mining, as well as genes linked to evolution, including those related to host adaptation [17,18]. *L. johnsonii* was suggested to be positively associated with the ability to resist chronic heat stress-induced diarrhea in weaned piglets, as revealed by comparative microbiome profiling analysis in our previous study [19]. Then, a total of 49 *L. johnsonii* strains were isolated from the feces of these heat-stress-tolerant piglets. A representative *L. johnsonii* strain, N5, was selected for in vivo experiments, and this strain was demonstrated to exhibit protective effects against dextran sulfate sodium (DSS)-induced colitis of the host by strengthening intestinal barrier function and balancing the response of intestinal innate and adaptive immunity [19]. In the current study, we conducted in vitro assessments of the antagonistic activity against pathogenic bacteria and the tolerance to gastric acid and bile of the *L. johnsonii* strains. Strain N5, along with another strain, N7, demonstrated notable antagonistic activity against *Salmonella* and exhibited high tolerance to acid and bile stresses. Complete genome sequences of strains N5 and N7 were determined by integrative analysis of Illumina short-read and nanopore long-read sequencing data. The genomic contents that potentially contributed to the intestinal colonization and functionality of *L. johnsonii* were identified. We further performed comparative genomic analyses of strains N5 and N7 with available complete genomes of *L. johnsonii* to discover shared and distinctive characteristics of N5 and N7, as well as to gain a comprehensive understanding of genome diversity and evolution of *L. johnsonii*.

## 2. Materials and Methods

### 2.1. Isolation of Lactic Acid Bacteria from Swine Feces

The LAB were isolated from the feces of weaned piglets that exhibited resistance to chronic heat stress (i.e., no diarrhea symptom) [19] through a dilution and plate coating method. Specifically, 2 g of each swine fecal sample was weighed and then added to a sterilized tube that contained 5 mL of sterilized PBS. The mixture was thoroughly mixed using vortex oscillation. The homogenized fecal samples were diluted to 10^−4^, 10^−5^, and 10^−6^, and 100 μL of each dilution were evenly spread on MRS plates, and incubated upside down at 37 °C for 24 h. Colonies were evenly distributed without any apparent adhesion phenomenon. A total of 30–300 colonies with different morphologies and colors were picked for subsequent cross-purification. Each strain was purified three times, followed by Gram staining and microscopic examination to ensure complete purification of the strain. In case of inconsistent morphology observed under microscopic examination, repeated line purification was performed until consistent morphology was achieved. The LAB was incubated in the corresponding medium at 37 °C for 18 to 24 h and subsequently sub-cultured under the same conditions for all subsequent assays.

### 2.2. Taxonomic Identification of LAB

The genomic DNA of each isolate was extracted using a bacterial genomic DNA extraction kit (Tiangen Biotech Co., Ltd. Beijing, China). Subsequently, the bacterial 16S rDNA sequence was amplified with the forward primer 27F (5′-AGAGTTTGATCCTGGCTCAG-3′) and reverse primer 1492R (5′-GGYTACCTTGTTACGACTT-3′). The PCR products were further verified through 1% agarose gel electrophoresis and subjected to Sanger sequencing (Genscript, Nanjing, China). The taxonomic affiliation of these strains was accomplished by comparing their sequences with 16S rDNA reference sequences deposited in the NCBI rDNA database using the BLASTn program. Out of the 57 obtained LAB strains, 49 strains were identified to be affiliated with *L. johnsonii* and subjected to further experiments.

### 2.3. Antagonistic Activity against Salmonella

The antibacterial activity was assessed following the methodology outlined by Leite et al. [20] with certain modifications. Ten microliters of each overnight *L. johnsonii* strain was spotted onto MRS agar and incubated for 24 h at 37 °C. Subsequently, the overnight culture of strain *Salmonella* (*S.*) *Typhimurium* SL1344 was inoculated into an LB semi-solid medium containing 0.8% agar (precooled to ambient temperature) at a ratio of 1:100 (*v*/*v*). The resulting mixture was evenly poured onto the MRS agar plates containing LAB strains. After solidification, the plates were incubated overnight at 37 °C. The presence of clear halo zones surrounding the *L. johnsonii* colony (measured in millimeters) was indicative of the antagonistic activity of the *L. johnsonii* strain. The strains that showed no evident halo information or a halo measuring less than 1 mm were recorded as “−”, and those that showed the presence of a distinct growth inhibition zone surrounding spots larger than 1 mm were scored as “+” (positive). Those with an inhibition zone ranging from 2 to 5 mm around the colony were recorded as “++”.

### 2.4. Bile and Acidic Tolerance

The *L. johnsonii* strains were cultured overnight in MRS broth at 37 °C and then inoculated into fresh MRS broth with a pH of 2.5 or 0.3% bovine bile salt at a 5% inoculation rate. After uniform distribution using a pipette, the strains were incubated at 37 °C for 4 h. A 0.1 mL aliquot of bacterial suspension was collected before culturing (0 h) and at two different time points (2 h and 4 h) for subsequent gradient dilution. Following appropriate dilutions (10^−3^, 10^−4^), a 0.1 mL diluent was applied onto MRS agar plates for enumeration. The tolerance rate of *L. johnsonii* strains to acidic and bile stresses (%) was determined by applying the following formula: (the number of viable bacteria at 2 and 4 h (CFU/mL)/the number of viable bacteria at 0 h (CFU/mL)) × 100%.

### 2.5. Analytical Profile Index (API) Experiment 

Single colonies of *L. johnsonii* N5 and N7 freshly cultured on MRS solid medium were selected and used to prepare a bacterial suspension with a turbidity of 2 McFarland in API 50 CHL medium (BioMérieux, Lyon, France). Subsequently, 5 mL of distilled water was poured into the honeycomb wells of the culture tray to create a humid environment. API 50 test strips were placed in the tray, and then bacterial suspension was added to each well of the biochemical test strip. The wells were then sealed with liquid paraffin and the tray was incubated at 37 °C for 24 h in a static incubator. Based on the instruction table, the biochemical reaction results were interpreted for N5 and N7.

### 2.6. Whole Genome Sequencing and Genome Assembly

The whole-genome sequencing was conducted on an Illumina HiSeq4000 platform (short reads) and a nanopore platform (long reads). High-quality long reads (with a Q score greater than 7 and length exceeding 1 kb) generated by the nanopore sequencing were assembled using Flye (v2.8.3) [21]. The generated assemblies were subsequently refined using NextPolish (v2) [22] with the Illumina-generated short reads as inputs. The genomic sequences of *L. johnsonii* N5 and N7 have been deposited at GenBank with the accession numbers CP136013 and CP136012. 

### 2.7. Comparative Genomics Analysis

Seventeen completely sequenced *L. johnsonii* genome sequences available in the NCBI database (as of 10 March 2022) (Table 1) were downloaded, and the genomes of N5 and N7 and 17 downloaded genomes were simultaneously annotated using Prokka (v1.14.6) [23]. The N5 and N7 strain-specific genes were identified using cd-hit-2d with parameters -s 0.9 and -c 0.9 [24]. The Clusters of Orthologous Groups of proteins (COG) annotations were obtained using emapper (v2.1.7) with default parameters, based on the eggNOG Roary orthology data (v5.0.2) [25], and the functional annotation was also completed by blasting genes against Kyoto Encyclopedia of Genes and Genomes (KEGG) databases [26]. The carbohydrate-active enzyme (CAZyme)-encoding genes in the *L. johnsonii* genomes were annotated using dbCAN (v2) with default parameters [27]. The average nucleotide identity (ANI) values between the *L. johnsonii* strains were determined using the recommended methods outlined in FastANI (v1.32) [28]. Roary (v3.13.0) was employed to analyze the pan- and core genome of the 19 *L. johnsonii* genomes [29].

## 3. Results and Discussion

### 3.1. Probiotic Properties of L. johnsonii Strains Isolated from Heat-Resistant Weaned Piglets

In our previous study, we found that *L. johnsonii* exhibited a relatively high abundance in the microbiota (>15%) of weaned piglets that were resistant to heat stress-induced diarrhea, while it was almost undetectable in the microbiota of those who had diarrhea [19]. The result suggested *L. johnsonii* probably exhibited probiotic effects to maintain the health of the host. A total of 49 *L. johnsonii* strains were subsequently isolated from the fecal samples of these heat-resistant piglets, and the beneficial properties of these strains were assessed. 

The possession of the capability to inhibit the growth of pathogens is crucial for evaluating the potential probiotic candidacy [30]. We assessed the antagonistic activity of these *L. johnsonii* strains using the strain *S. Typhimurium* SL1344, which represents one of the most prominent pathogens in animals, as an indicator. The results demonstrated that the antagonistic ability varied among these *L. johnsonii* strains (Appendix A), and two representative strains, N5 and N7, exhibited superior antagonistic effects compared with others. The tolerance to acidic and bile stresses of strains N5 and N7 were then assessed. This evaluation is pertinent because potential probiotic strains must demonstrate resistance against acidic and bile stresses in the gastrointestinal tract of animals to facilitate colonization. In an acidic environment with a pH of 2.5, the survival rate of *L. johnsonii* N5 and N7 were 110.34% ± 5.97% (mean ± SD) and 73.33% ± 1.53% after 2 h, and 65.52% ± 6.65% and 33.33% ± 2.25% after 4 h, respectively (Figure 1A). In a 0.3% ox gall solution, the survival rate of *L. johnsonii* N5 and N7 were 95.74% ± 6.38% (mean ± SD) and 92.31% ± 5.77% after 2 h, and 76.60% ± 6.38% and 113.46% ± 8.81% after 4 h, respectively (Figure 1B). The results demonstrated that the growth activity of N5 and N7 in the acidic and bile environments gradually declined over time, while still maintaining a relatively high survival rate. Then, strain N5 was selected for in vivo probiotic capacity assessment using a mouse model, and the results demonstrated that oral administration of strain N5 alleviated the clinical and histological signs of colitis of the host through several means, including suppressing the expression of pro-inflammatory cytokines TNF-α and IL-6 within the intestinal tract [19]. 

### 3.2. General Genomic Features of the L. johnsonii N5 and N7 Strains

To identify potential genomic contents that were associated with the observed probiotic properties as mentioned above, the complete genome sequences of strains *L. johnsonii* N5 and N7 were determined by a combination of Illumina and nanopore sequencing technologies. A single circular chromosome 1,966,342 bp (Figure 2A) and 1,966,349 bp (Figure 2B) was generated for strains N5 and N7, respectively, with no plasmid identified in either strain. The whole genome ANI value between the two strains was 99.998%, as revealed by FastANI analysis. Then, genes were predicted from the two genomes using PROKKA, and 1863 and 1870 CDSs were identified in N5 and N7. With a threshold of similarity of 90% and coverage of 90% using cd-hit-2d, only two N5-specific and four N7-specific genes were identified, with the remaining genes conserved between the two strains. These results demonstrated that the two strains shared very high genomic contents. The two N5-specific genes included one gene annotated as *typA* and another annotated as a gene encoding a hypothetical protein, and the four N7-specific genes included two IS30 family transposase ISLga1-encoding genes and two hypothetical protein-encoding genes. *TypA* encodes a ribosome-binding GTPase that is necessary for survival under some stress conditions and contributes to rapid attachment and biofilm formation [31]. 

COG and KEGG functional annotations were assigned to the commonly shared genes of the two strains. A total of 1737 CDSs obtained functional annotations that were specifically assigned to 20 COG categories (Figure 3A). The most prevalent COG term was “function unknown” (351 genes, 20.21%), and the remaining genes were primarily classified into functional categories associated with replication, recombination, and repair (190 genes, 10.94%), translation, ribosomal structure and biogenesis (160 genes, 9.21%), carbohydrate transport and metabolism (145 genes, 8.35%), transcription (143 genes, 8.23%), amino acid transport and metabolism (114 genes, 6.56%), cell wall/membrane/envelope biogenesis (99 genes, 5.70%), and nucleotide transport and metabolism (89 genes, 5.12%). Furthermore, a total of 1087 CDSs were assigned to 40 KEGG categories (Figure 3B), mainly functioning in the metabolic pathways (231 genes, 21.25%), biosynthesis of secondary metabolites (86 genes, 7.91%), ABC transporters (53 genes, 4.88%), ribosome (53 genes, 4.88%), microbial metabolism in diverse environments (47 genes, 4.32%), and biosynthesis of cofactors (32 genes, 2.94%).

### 3.3. Stress Resistance-Associated Genes

Several genes associated with resistance to the acidic and bile stresses were identified in the *L. johnsonii* N5 and N7 genomes. For instance, *cbh*, which encodes a chenodeoxycholylglycine hydrolase, and *nhaC*, which encodes a Na^+^:H^+^ antiporter, were identified in both genomes. *Cbh* and *nhaC* are found to be essential for the survival of *L. johnsonii* in acidic and bile salt environments [32]. Two choloylglycine hydrolase-encoding genes were identified in the N5 and N7 genomes. Choloylglycine hydrolase is involved in adaptation to the bile stress and recognized as one of the gut-specific genes of LAB that colonize the intestines [33]. Exopolysaccharides (EPS) produced by LAB play a role in conferring resistance to environmental stresses, including bile salts and acidic pH within the gastrointestinal tract. Additionally, these EPS facilitate the attachment of LAB to intestinal cells in the host, preventing competing pathogenic bacteria from attaching to the host cells [34,35,36]. A generic EPS cluster was identified in strains N5 and N7. Interestingly, in the cluster, *epsABCD* exhibited very high similarity with the well-studied strain *L. johnsonii* FI9785 [37] (>90% protein sequence identity for all the four genes); however, *epsE* exhibited relatively low similarity (protein sequence identity 44.23%) with that of FI9785 (FI9785_1179) and showed high similarity (94.67%) with that of strain MR1, indicative of occurrence of recombination events in the EPS cluster of strains N5 and N7.

Furthermore, several genes associated with resistance to heat stress, including a master transcriptional regulator encoding gene *cspA* and several genes encoding chaperone-activated proteins, such as HtpX, Hsp20, Hsp33, GroEL, GroES, HslO, and DnaK [38,39], were identified in the *L. johnsonii* N5 and N7 genomes. Several genes involved in resistance against oxidative stress, such as *trxB* (encoding a thioredoxin reductase) and *msrAB* (encoding a peptide–methionine (S)-S-oxide reductase) [38,40], were also identified in the *L. johnsonii* N5 and N7 genomes. The expression of *trxB* in *L. plantarum* WCFS1 significantly enhanced the strain’s resistance to oxidative stress induced by hydrogen peroxide or diamide [41]. *HslV*, whose product is implicated in stress response associated with the maintenance of protein homeostasis and prevention of protein damage [5], was identified in the *L. johnsonii* N5 and N7 genomes. Furthermore, the *L. johnsonii* N5 and N7 genomes were found to harbor 4 sigma factors and 20 transcriptional regulators involved in transcriptional regulation, thereby adequately fulfilling the transcriptional regulatory requirements of *L. johnsonii* in a nutrient-rich but stressful gut environment. Overall, the presence of these stress resistance-related genes confers enhanced adaptability to gastrointestinal stress conditions in the *L. johnsonii* N5 and N7 genomes, thereby assisting the improved survival and proliferation of *L. johnsonii* in response to environmental stresses in the gastrointestinal tract. 

### 3.4. Anti-Inflammation-Associated Genes

Inflammation is a natural response to pathogen infection, but prolonged or uncontrolled inflammation can lead to tissue damage. Our previous study demonstrated that oral administration of *L. johnsonii* N5 could reduce the expression level of proinflammatory cytokine TNF-α in the intestines [19]. Several anti-inflammation-associated genes were identified in both the N5 and N7 genomes. For instance, *gro*EL and *gro*ES were identified in both genomes. The GroEL/GroES chaperone system is responsible for correctly folding proteins in an ATP-regulated manner [42], and GroEL of *Lactobacillus* has been demonstrated to exhibit an immunomodulatory effect and can inhibit the production of the proinflammatory cytokine TNF-α induced by LPS [43]. Additionally, the N5 and N7 genomes were found to harbor *folC*, which is involved in folate metabolism and can regulate histamine production. Previous studies suggested that histamine derived from *Lactobacillus reuteri* can inhibit intestinal inflammation by regulating proinflammatory cytokines through PKA and ERK signaling [44]. Two key genes involved in the biosynthesis of capsular polysaccharides (CPS), *capA* and *capB*, were identified in the N5 and N7 genomes. The CPS has been reported to promote macrophage proliferation and the production of regulatory T (Treg) cells in the intestine, which can affect the balance between Tregs and Th17 cells and ultimately prevent the development of colitis [45]. Furthermore, EPS is also believed to have functions in immunomodulation [46]. The presence of the above-mentioned anti-inflammatory genes probably enables *L. johnsonii* to cope well with intestinal inflammation and protect host health.

### 3.5. CAZymes

CAZymes are ubiquitously present in various organisms, with a particularly high abundance observed in microorganisms [40,47]. In this study, CAZymes were annotated from *L. johnsonii* N5 and N7 genomes using dbCAN2, and an identical number of CAZymes were identified in N5 and N7 genomes, including three carbohydrate esterase (CE) families, 13 carbohydrate-binding modules (CBM) families, 57 glycosyl transferases (GT) families, and 62 glycoside hydrolases (GH) families (Figure 4). Genes belonging to GHs and GTs exhibited the highest number in *L. johnsonii* N5 and N7, which is consistent with that identified in other *Lactobacillus* species including *L. salivarius* [17]. The GHs play a crucial role in the degradation of glycosidic bonds between the chain residues of the carbohydrate compounds, with GH13 being the predominant subgroup in *L. johnsonii* N5 and N7. GH13 is essential for the degradation of various carbohydrates including amylase, β-glucosidase, β-xylosidase, and β-galactosidase [48]. GTs facilitate the formation of glycosidic bonds by transferring sugar residues from donors to acceptors, encompassing carbohydrates, proteins, lipids, DNA, and other biomolecules [49,50]. GT2 and GT4 were the most predominant GT subgroups in *L. johnsonii* N5 and N7. Generally, GT2 is responsible for the biosynthesis of cellulose, undecaprenyl-diphospho-muramoyl pentapeptide, beta-N-acetylglucosaminyltransferase, and putative glycosyltransferase, while GT4 is responsible for facilitating the activity of the Sec-dependent glycosyltransferase (*gtfA)*, which plays a crucial role in sucrose biosynthesis [51,52]. Overall, these genes actively participate in the metabolism and transportation of functional and bioactive substances, thereby facilitating efficient nutrient transport and synthesis by *L. johnsonii* N5 and N7 within the intestinal tract.

### 3.6. Biosynthesis and Transport System

Both N5 and N7 harbored *asnA* and *asnB*, encoding asparagine synthetases. Asparagine synthetases are involved in the synthesis of L-asparagine from L-aspartic acid, contributing to efficient ammonia assimilation [53]. Additionally, both strains also carried *glnA*, whose product is a glutamine synthetase (GLUL) that catalyzes the formation of glutamine by combining glutamic acid and ammonia. The presence of glycine hydroxymethyltransferase (SHMT) encoded by *glyA* suggested that *L. johnsonii* N5 and N7 can facilitate the interconversion between serine and glycine. Moreover, *L. johnsonii* N5 and N7 could synthesize lysine from aspartic acid through a series of enzymes encoded by *lysC*, *asd*, *dapA*, *dapB*, *dapH*, *patA*, *dapL*, *dapF*, and *lysA*. Furthermore, *L. johnsonii* N5 and N7 exhibited robust transport systems, as evidenced by the presence of a total of 70 genes associated with transport mechanisms in their genomes. Notably, these included 53 ABC transport system genes that are responsible for the efficient transportation of amino acids and peptides, thereby facilitating protein synthesis (Appendix A). The genomes of *L. johnsonii* N5 and N7 also contained 17 genes associated with the phosphoenolpyruvate-dependent sugar phosphotransferase system (PTS) including PTS enzyme I (PTS-EI) and PTS-EII complexes (Appendix A), which catalyze the phosphorylation of carbohydrate substrates across the microbial cell membrane and serve as a prominent active carbohydrate transport mechanism in bacteria [54]. Specifically, the *L. johnsonii* N5 and N7 genomes carried *ptsI* and *ptsH*, which encoded PTS-EI and the phosphor carrier protein HPR, respectively. The two genes collaborate to facilitate the transfer and transport of phosphorylated compounds by transferring phosphoenolpyruvate to PTS-EII [5,38]. The genomes of N5 and N7 encoded 14 PTS-EII complexes, surpassing the count observed in other microorganisms with comparable genome sizes [53]. The information of these complexes is closely linked to the transport of diverse carbon sources, including glucose, β-glucosides, disaccharides, mannitol, sorbitol, mannose, and sucrose [55]. To validate the predictions on carbohydrate metabolism, we conducted Analytical Profile Index (API) identification using a rapid API-50 CH biochemical test kit. The biochemical activities of D-galactose, D-glucose, salicin, D-cellobiose, D-maltose, D-melibiose, D-sucrose, D-trehalose, mannitol, methyl-β-D-fucopyranoside, and methyl-α-D-glucopyranoside are positive (Table 2), further suggesting that these PTSs enable the introduction of multiple substrates and facilitate the transport of diverse sugars, thus augmenting the carbon transport capacity of *L. johnsonii* N5 and N7 genomes. 

### 3.7. Genome Comparison of Strains N5 and N7 and Related Complete L. johnsonii Genomes

Comparative genomic analyses were performed using our newly sequenced N5 and N7 genomes and 17 complete *L. johnsonii* genomes available in the NCBI Refseq database. ANI was used to assess genomic relatedness between these *L. johnsonii* strains [56,57]. The commonly employed threshold of ANI values for demarcating bacterial species based on their genomic sequence is 95% [28]. In this study, the ANI values of the 19 *L. johnsonii* strains were all higher than 95% (Appendix A), indicating that these strains were affiliated with the same species. A phylogenetic tree based on the core genome sequences was constructed to enhance our understanding of the phylogenetic relationships among strains of *L. johnsonii*. The phylogenetic tree was classified into three distinct clusters (Figure 5), and *L. johnsonii* N5 and N7 were found to be closely associated with GHZ10a and ZLJ010, which were isolated from swine feces [5]. 

The core genome and pan-genome are commonly employed for evaluating the genomic diversity of a species or closely related bacteria [58]. The core genome of a species encompasses the complete set of genes shared by all strains, including the genetic determinants that maintain species-specific characteristics; on the other hand, the pan-genome represents the collective gene pool that can be accommodated by the genetic determinants, including core genes and accessory genes (i.e., genes not harbored by all strains) [59,60]. The pan-genome of *L. johnsonii* comprised 5977 genes across 19 strains, and its expansion with the addition of newly sequenced genomes signified an open nature of the *L. johnsonii* pan-genome (Figure 6A). The estimated number of newly added genes per genome sequence revealed a rapid decline in the average count from the second genome sequence onwards, interspersed with occasional fluctuations indicating sporadic increases in subsequent genomes (Figure 6B). Even in the 19th genome, novel genes continued to be incorporated, suggesting ongoing gene exchange within *L. johnsonii* species and between other bacterial species, with a predominant emergence of new functionalities during evolutionary processes. This observation may be attributed to the extensive genetic variability among strains, enabling them to effectively adapt to the host intestinal environment, and it is plausible that these phenomena could also be associated with the insertion and translocation of insertion sequences (IS), thereby facilitating gene recombination [61]. These accessory genes could contribute to bacterial evolution and adaptation to diverse environments [62]. In contrast to the pan-genome, the core genome of *L. johnsonii* contained 1044 genes and exhibited gradual stabilization as the number of genomes increased to 15. Consequently, an increase in the number of genomes exerted a relatively minor influence on the magnitude of the core genome. 

### 3.8. Unique Genomic Characteristics of L. johnsonii N5 and N7

The *L. johnsonii* N5 and N7 genomes collectively harbor a total of 115 unique genes, compared with the other 17 complete *L. johnsonii* genomes (Appendix A), and 99 out of these 115 genes showed homology with genes deposited in the NCBI nr database; 16 genes were identified as novel genes based on the BLAST analysis (identity threshold 80% and coverage threshold 80%). The potential function of these N5- and N7-specifically harbored genes was then predicted, 86 genes obtained COG annotation and 29 genes were identified as hypothetical proteins with undetermined functions. The 86 COG annotated genes were assigned to 17 COG categories, with “replication, recombination, and repair” (L) being the most prevalent category (including 45 genes, accounting for 52.3% of the 86 annotated genes). Among the 45 genes assigned to COG category L, 37 genes encode transposases, and 20 of these encode transposases belonging to the IS30 family, which can enable the insertion or removal of DNA fragments at new positions in the genomes [40,63]. The process of gene recombination can enhance the diversity and adaptability of *L. johnsonii* in diverse environments, thereby conferring advantageous traits for survival and reproduction within the host’s surrounding milieu [63]. Furthermore, nine genes belonged to the “transcription” category (K), including several transcriptional regulators. Among them, one gene was annotated to encode a transcriptional factor AsnR. AsnR plays an important role in regulating the utilization of asparagine by the bacterial host [64]. A RimI encoding gene was also identified in the genes belonging to COG category K. RimI is demonstrated to be important in regulating the growth rate of bacterial cells by modification of bacterial translation apparatus [65]. These regulators possibly exert their influence on specific genes, thereby governing gene expression and fulfilling crucial regulatory functions in bacterial metabolism, stress response, and other physiological processes, enabling the strain to acquire advantageous traits within the intestinal tract [66]. The “cell wall/membrane/envelope biogenesis” category (M) encompassed nine genes, primarily encoding GTs, and these GTs are closely associated with EPS synthesis [67]. Previous research has demonstrated that GTs, particularly glucosyltransferase (GTF), are essential for EPS biosynthesis [68,69]. Therefore, it is postulated that the presence of these specific GTs confers a competitive advantage to *L. johnsonii* N5 and N7 in the host’s intestinal environment. Of note, a large fraction of the N5- and N7-specifically harbored genes lacked definite functional annotations (Appendix A). The functions of these genes need to be further determined by future experiments, such as gene deletion-, transcriptomics-, and proteomics-based approaches. 

## 4. Conclusions

Considering the plethora of probiotic properties of *L. johnsonii* N5 and N7, these strains could be considered as potential probiotic candidates. The complete genome sequences of the two strains were obtained by the integrative combination of short-read Illumina and long-read nanopore sequencing. The two strains harbored highly conserved genomic backgrounds, with two N5-specific and four N7-specific genes identified. Several genes that were involved in adaptation in the gastrointestinal environments and probiotic properties including anti-inflammation were identified in both genomes. Through comparative genomics analysis with 17 available complete *L. johnsonii* genomes, 115 genes that were specifically harbored by N5 and N7 were identified, which mainly encoded proteins involved in transposase, replication, recombination and repair, and transcription. The presence of genes with probiotic properties such as those involved in resistance against acidic and bile stresses, anti-inflammation, and CAZymes, probably enables N5 and N7 to efficiently adapt to the host environment and exert beneficial effects. In summary, the genome-wide analysis of the two *L. johnsonii* strains provides further insight into the niche adaptation and probiotic properties of *L. johnsonii* and provides clues for targeted experimental studies.

## Figures and Tables

**Figure 1 microorganisms-12-00672-f001:**
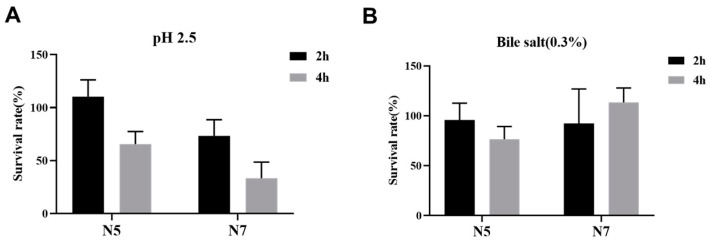
Survival rates of *L. johnsonii* N5 and N7 under pH 2.5 (**A**) and bile salt (**B**) stresses. Data shown are mean ± SD of triplicate values of independent experiments.

**Figure 2 microorganisms-12-00672-f002:**
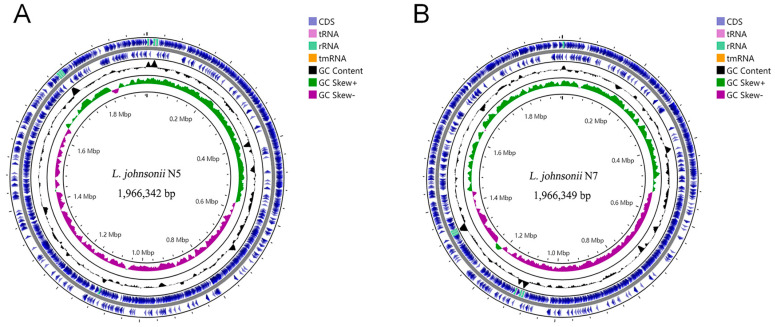
Genome atlases of *L. johnsonii* N5 (**A**) and *L. johnsonii* N7 (**B**). Each side of the figure consists of five circles. From the outermost circle to the inside, circles 1 and 2 show the distribution of coding genes in the front chain and the back chain, including tRNA (purple) and rRNA (green), circle 3 shows the GC content, and circles 4 and 5 show GC Skew^+^ and GC Skew^-^ respectively.

**Figure 3 microorganisms-12-00672-f003:**
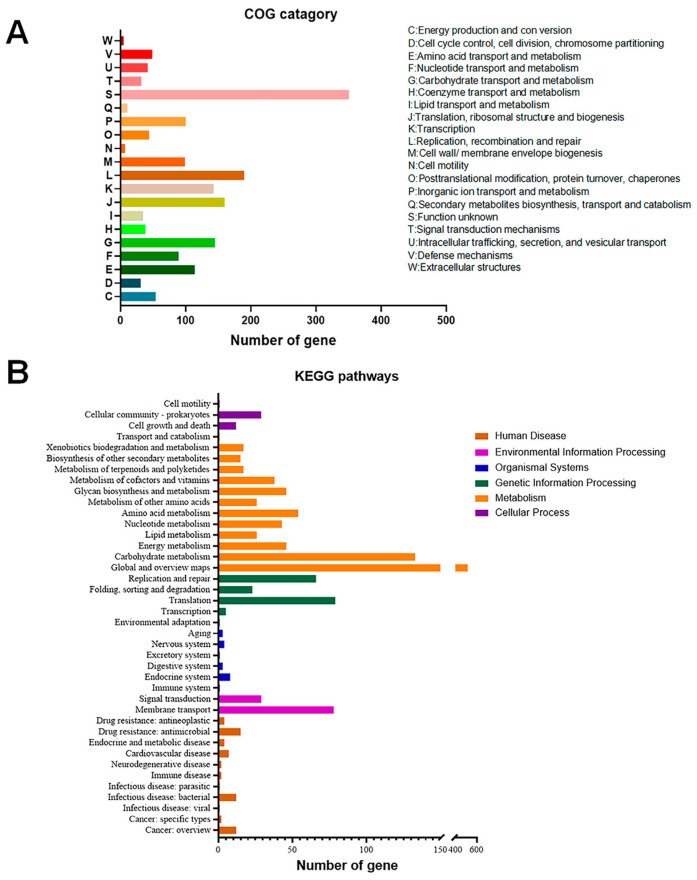
The number of genes assigned to COG (**A**) and KEGG (**B**) categories. COG, Clusters of Orthologous Group; KEGG, Kyoto Encyclopedia of Genes and Genomes.

**Figure 4 microorganisms-12-00672-f004:**
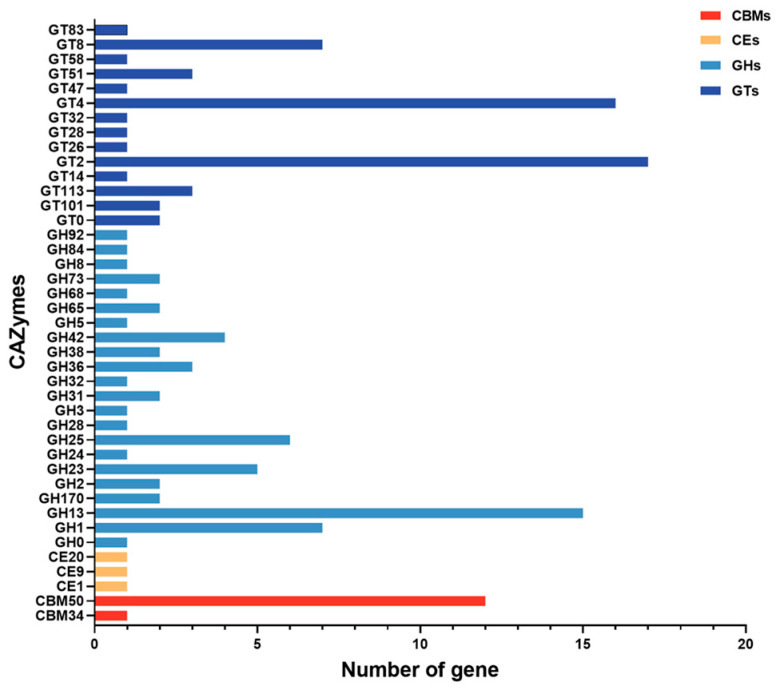
Distribution of CAZymes in the *L. johnsonii* N5 and N7 genomes.

**Figure 5 microorganisms-12-00672-f005:**
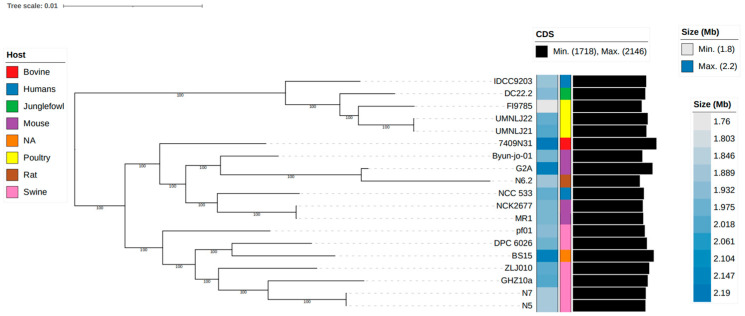
Phylogenetic relationship between representative *L. johnsonii* genomes and strains N5 and N7 as revealed by Roary analysis, and displayed using iTOL. The genome size, origin, and number of CDSs are listed alongside the strain name.

**Figure 6 microorganisms-12-00672-f006:**
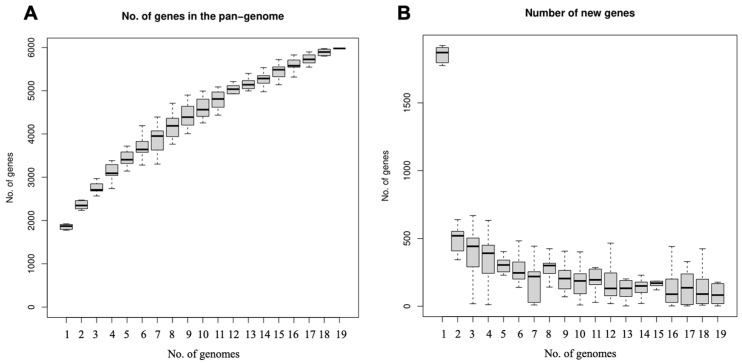
*L. johnsonii* N5 and N7 pan-genome (**A**). Number of new genes identified in *L. johnsonii* with sequential addition of new genomes (**B**).

**Table 1 microorganisms-12-00672-t001:** Summary of *Lactobacillus johnsonii* genomes used in this study.

Strain	RefSeq Assembly Accession	GC%	Size (Mb)	CDS	Plasmid	Isolation Source	Host	Country
N5 (this study)	GCF_032463685.1	35	1.88	1863	0	Feces	Swine	China
N7 (this study)	GCF_032463545.1	35	1.88	1870	0	Feces	Swine	China
GHZ10a	GCF_014841035.1	34.9	2.01	1921	2	Feces	Swine	China
7409N31	GCF_022810665.1	35	2.19	2146	0	Feces	Bovine	NA
BS15	GCF_001714745.1	34.9	2.16	2076	1	Yoghurt	-	China
ZLJ010	GCF_004011315.1	34.9	2.00	1959	0	Intestine	Swine	China
UMNLJ22	GCF_002176835.1	34.6	1.99	1922	2	Ileum	Poultry	USA
UMNLJ21	GCF_002176855.1	35.12	2.01	1888	2	Ileum	Poultry	USA
Byun-jo-01	GCF_003316915.1	34.7	1.96	1781	0	Jejunum	Mouse	South Korea
MR1	GCF_022213385.1	34.7	1.95	1805	0	Mice cecum	Mouse	USA
NCK2677	GCF_014058685.1	34.7	1.95	1792	0	Mouse	Mouse	USA
G2A	GCF_010586925.1	34.6	2.17	2045	2	Mouse	Mouse	USA
DC22.2	GCF_009769185.1	34.5	1.94	1858	4	Birds	Junglefowl	UK
IDCC9203	GCF_003428395.1	34.7	1.90	1882	0	Infant feces	Humans	South Korea
NCC 533	GCF_000008065.1	34.6	1.99	1821	0	Humans	Humans	NA
DPC 6026	GCF_000204985.1	34.8	1.97	1900	0	Small intestine	Swine	NA
N6.2	GCF_000498675.1	34.2	1.89	1718	0	Rat	Rat	NA
FI9785	GCF_000091405.1	34.5	1.76	1765	2	Poultry	Poultry	UK
pf01	GCA_000219475.3	34.6	1.93	1846	2	Piglet feces	Swine	Korea

Note: the genomes of strains N5 and N7 were determined in this study. ‘NA’ denotes no exact geographic information was obtained based on the information listed in the NCBI Biosample database.

**Table 2 microorganisms-12-00672-t002:** Carbohydrate fermentation profile of *L. Johnsonii* N5 and N7 as assessed by API 50 CH test strips.

Substrate	Result		Substrate	Result	
	N5	N7		N5	N7
Control	−	−	Esculine	−	−
Glycerol	−	−	Salicine	+	+
Erythritol	−	−	D-cellobiose	+	+
D-arabinose	−	−	D-maltose	+	+
L-arabinose	−	−	D-lactose	−	−
D-ribose	−	−	D-melibiose	+	+
D-xylose	−	−	D-saccharose	+	+
L-xylose	−	−	D-threalose	−	−
D-adonitol	−	−	Inulin	−	−
Methyl-β-dxylopyranoside	+	+	D-melezitose	+	+
D-galactose	+	+	D-raffinose	−	−
D-glucose	+	+	Starch	−	−
D-fructose	−	−	Glycogene	−	−
D-mannose	−	−	Xylitol	+	+
L-sorbose	−	−	Gentiobiose	−	−
L-rhamnose	−	−	D-turanose	−	−
Dulcitol	−	−	D-lyxose	−	−
Inositol	−	−	D-tagatose	−	−
D-mannitol	−	−	D-fucose	−	−
D-sorbitol	−	−	L-fucose	−	−
Methyl-αD-mannopyranoside	−	−	D-arabitol	−	−
Methyl-αD-glucopyranoside	+	+	L-arabitol	−	−
N-acetylglucosamine	−	−	Potassium gluconate	−	−
Amygdaline	−	−	Potassium 2-cetogluconate	−	−
Arbutine	+	+	Potassium 5-cetogluconate	−	−

Notes: “−” denotes the substrate cannot be utilized, and “+” denotes the substrate can be utilized.

## Data Availability

The genomic sequences of *L. johnsonii* N5 and N7 have been deposited at GenBank with the accession numbers CP136013 and CP136012.

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
