# Peer review of "Characterization of Probiotic Properties and Whole-Genome Analysis of Lactobacillus johnsonii N5 and N7 Isolated from Swine"

_microorganisms, 2024, doi:10.3390/microorganisms12040672_

Round 1

Reviewer 1 Report

Comments and Suggestions for Authors

This is an interesting and well written paper, reporting and discussing the probiotic characteristics of two L. johnsonii N5 and N7 that developed interesting properties, including high resistance under acidic and bile environment conditions and antagonism to Salmonella, as well as the results of genomic analysis for elucidating and correlating such probiotic properties with genetic basis of adaptation and genome diversity. The methodology based on whole-genome investigation was well conceived, highly detailed, and interpreted.

As suggestion for increasing the value of the manuscript, it would be interesting to mention and discuss a little bit the other omic approach based on proteomics, which is also very helpful to identify, compare, and correlate functional genes to the signature protein profiles to the probiotic strain functionalities.  

The manuscript deserves to be published after possibly minor revision.

Author Response

Thanks for your encouragement and valuable comments. Based on our suggestion, we added discussion in line 436 “Of note, a large fraction of the N5- and N7-specifically harbored genes lacked definite functional annotations (Table S3). The functions of these genes need to be further determined by future experiments, such as gene deletion-, transcriptomics-, and proteomics-based approaches.”. Hopefully, our revised manuscript can address your concerns. Thanks again for your help in improving the manuscript.

Reviewer 2 Report

Comments and Suggestions for Authors

The aims of the study are clearly presented both in the abstract and at the end of the introduction.

The authors describe the core and pan-genomes of the two Lactobacillus johnsonii strains in some detail. Moreover, they give a summarised descriptive analysis of the 101 unique genes harboured by these two strains which are missing in all others. However, they fail to explain the potential function of the two N5-specific and four N7-specific genes identified. Could these be related to their exhibited probiotic properties? What are these genes doing? Which proteins are they coding for? Are these orphan genes? Do they exist, for example, in other species with probiotic potential?

The authors stated in the objectives of their study that they sequenced these strains to "...elucidate the genetic basis of these observed probiotic properties....", but it seems that they fail to do so.

In their "Conclusions" section the authors mention "The presence of genes with probiotic properties...". Which are these genes? What are they responsible for exactly? Which probiotic properties?

This analysis is missing and would constitute the distinctive and most interesting part of the present study!

Minor comments:

line 98: please use "Gram staining"

Higher definition images are needed at least for Figures 2 and 3.

Author Response

The authors describe the core and pan-genomes of the two Lactobacillus johnsonii strains in some detail. Moreover, they give a summarised descriptive analysis of the 101 unique genes harboured by these two strains which are missing in all others. However, they fail to explain the potential function of the two N5-specific and four N7-specific genes identified. Could these be related to their exhibited probiotic properties? What are these genes doing? Which proteins are they coding for? Are these orphan genes? Do they exist, for example, in other species with probiotic potential?

R: Thanks a lot for your valuable comments. Based on your comments, we performed a further more detailed analysis on these unique genes harbored by strains N5 and N7. Through carefully manual curation, the unique gene set was updated to harbor 115 genes, and the Table S3 was updated accordingly. These genes were blasted against the NCBI nr database, and the results demonstrated that 99 out of these 115 genes showed homology with genes deposited in the NCBI nr database, with 16 genes identified as novel genes based on the BLAST analysis (identity threshold 80% and coverage threshold 80%). The potential function of these N5- and N7-specifically harbored genes was then predicted, 86 genes obtained COG annotation and 29 genes were identified as hypothetical proteins with undetermined functions. We then performed a literature search to identify genes with definite functions as revealed by previous studies in other bacterial species, and found several genes were involved in host adaptation and probiotic traits (lines 413-435). A large fraction of the unique genes harbored by strains N5 and N7 did not obtain definite functional annotation. We also added discussion line 436-439 “Of note, a large fraction of the N5- and N7-specifically harbored genes lacked definite functional annotations (Table S3). The functions of these genes need to be further determined by future experiments, such as gene deletion-, transcriptomics-, and proteomics-based approaches”. Thanks again for your valuable comments. Below please find our revised paragraph

“The L. johnsonii N5 and N7 genomes collectively harbor a total of 115 unique genes compared with the other 17 complete L. johnsonii genomes (Table S3), and 99 out of these 115 genes showed homology with genes deposited in the NCBI nr database, with 16 genes identified as novel genes based on the BLAST analysis (identity threshold 80% and coverage threshold 80%). The potential function of these N5- and N7-specifically harbored genes was then predicted, 86 genes obtained COG annotation and 29 genes were identified as hypothetical proteins with undetermined functions. The 86 COG annotated genes were assigned to 17 COG categories, with “replication, recombination, and repair” (L) being the most prevalent category (including 45 genes, accounting for 52.3% of the 86 annotated genes). Among the 45 genes assigned to COG category L, 37 genes encode transposases, and 20 out of them encoded transposases belonging to the IS30 family, which can enable the insertion or removal of DNA fragments at new positions in the genomes [40,64]. The process of gene recombination can enhance the diversity and adaptability of L. johnsonii in diverse environments, thereby conferring advantageous traits for survival and reproduction within the host's surrounding milieu [64]. Furthermore, nine genes belonged to the “transcription” category (K), including several transcriptional regulators. Among them, one gene was annotated to encode a transcriptional factor AsnR. AsnR plays an important role in regulating the utilization of asparagine by the bacterial host [65]. A RimI encoding gene was also identified in the genes belonging to COG category K. RimI is demonstrated to be important in regulating the growth rate of bacterial cells by modification of bacterial translation apparatus [66]. These regulators possibly exert their influence on specific genes, thereby governing gene expression and fulfilling crucial regulatory functions in bacterial metabolism, stress response, and other physiological processes, enabling the strain to acquire advantageous traits within the intestinal tract [67]. The “cell wall/membrane/envelope biogenesis” category (M) encompassed nine genes, primarily encoding GTs, and these GTs are closely associated with EPS synthesis [68]. Previous research has demonstrated that GTs, particularly glucosyltransferase (GTF), are essential for EPS biosynthesis [69,70]. Therefore, it is postulated that the presence of these specific GTs confers a competitive advantage to L. johnsonii N5 and N7 in the host's intestinal environment. Of note, a large fraction of the N5- and N7-specifically harbored genes lacked definite functional annotations (Table S3). The functions of these genes need to be further determined by future experiments, such as gene deletion-, transcriptomics-, and proteomics-based approaches.”

The authors stated in the objectives of their study that they sequenced these strains to "...elucidate the genetic basis of these observed probiotic properties....", but it seems that they fail to do so.

R: Thanks for your valuable comment. This sentence has been revised as “To identify genes potentially involved in these observed probiotic properties, the complete genome sequences of N5 and N7 were determined using a combination of Illumina and Nanopore sequencing.”

In their "Conclusions" section the authors mention "The presence of genes with probiotic properties...". Which are these genes? What are they responsible for exactly? Which probiotic properties? This analysis is missing and would constitute the distinctive and most interesting part of the present study!

 R: Thanks for your valuable comment. In this study, we found strains L. johnsonii N5 and N7 exhibited probiotic properties, and identified several genes involved in these observed probiotic properties, such as resistance against acidic and bile stresses, anti-inflammation, and CAZymes. The genes involved in these probiotic properties were described and discussed in the main text. Furthermore, several genes specifically harbored by strains N5 and N7 but not identified in the other 17 L. johnsonii strains were also suggested to be involved in host adaptation and probiotic properties, and were described in the revised manuscript (line 416-436). Based on your suggestion, the sentence in the conclusion section has been revised as “The presence of genes with probiotic properties such as those involved in resistance against acidic and bile stresses, anti-inflammation, and CAZymes, probably enables N5 and N7 to efficiently adapt to the host environment and exert beneficial effects.”

The potential function of unique genes specifically harbored by strains N5 and N7 were also discussed as follows: “The L. johnsonii N5 and N7 genomes collectively harbor a total of 115 unique genes compared with the other 17 complete L. johnsonii genomes (Table S3), and 99 out of these 115 genes showed homology with genes deposited in the NCBI nr database, with 16 genes identified as novel genes based on the BLAST analysis (identity threshold 80% and coverage threshold 80%). The potential function of these N5- and N7-specifically harbored genes was then predicted, 86 genes obtained COG annotation and 29 genes were identified as hypothetical proteins with undetermined functions. The 86 COG annotated genes were assigned to 17 COG categories, with “replication, recombination, and repair” (L) being the most prevalent category (including 45 genes, accounting for 52.3% of the 86 annotated genes). Among the 45 genes assigned to COG category L, 37 genes encode transposases, and 20 out of them encoded transposases belonging to the IS30 family, which can enable the insertion or removal of DNA fragments at new positions in the genomes [40,64]. The process of gene recombination can enhance the diversity and adaptability of L. johnsonii in diverse environments, thereby conferring advantageous traits for survival and reproduction within the host's surrounding milieu [64]. Furthermore, nine genes belonged to the “transcription” category (K), including several transcriptional regulators. Among them, one gene was annotated to encode a transcriptional factor AsnR. AsnR plays an important role in regulating the utilization of asparagine by the bacterial host [65]. A RimI encoding gene was also identified in the genes belonging to COG category K. RimI is demonstrated to be important in regulating the growth rate of bacterial cells by modification of bacterial translation apparatus [66]. These regulators possibly exert their influence on specific genes, thereby governing gene expression and fulfilling crucial regulatory functions in bacterial metabolism, stress response, and other physiological processes, enabling the strain to acquire advantageous traits within the intestinal tract [67]. The “cell wall/membrane/envelope biogenesis” category (M) encompassed nine genes, primarily encoding GTs, and these GTs are closely associated with EPS synthesis [68]. Previous research has demonstrated that GTs, particularly glucosyltransferase (GTF), are essential for EPS biosynthesis [69,70]. Therefore, it is postulated that the presence of these specific GTs confers a competitive advantage to L. johnsonii N5 and N7 in the host's intestinal environment. Of note, a large fraction of the N5- and N7-specifically harbored genes lacked definite functional annotations (Table S3). The functions of these genes need to be further determined by future experiments, such as gene deletion-, transcriptomics-, and proteomics-based approaches.”

Minor comments:

line 98: please use "Gram staining"

R: corrected, thanks.

Higher definition images are needed at least for Figures 2 and 3.

R: the quality of all figures has been improved and the resolution is at least 300 dpi. Thanks.

Many thanks for your help in improving the manuscript. Hopefully, our revised manuscript can address your concerns.

Round 2

Reviewer 2 Report

Comments and Suggestions for Authors

The authors have satisfactorily replied to my questions and revised the manuscript accordingly.